# Steroidogenic Factor 1, a Goldilocks Transcription Factor from Adrenocortical Organogenesis to Malignancy

**DOI:** 10.3390/ijms24043585

**Published:** 2023-02-10

**Authors:** Lauriane Relav, Mabrouka Doghman-Bouguerra, Carmen Ruggiero, João C. D. Muzzi, Bonald C. Figueiredo, Enzo Lalli

**Affiliations:** 1Institut de Pharmacologie Moleculaire et Cellulaire CNRS UMR 7275, 06560 Valbonne, France; 2Universite Cote d’Azur, 06560 Valbonne, France; 3Laboratório de Imunoquímica (LIMQ), Pós-Graduação em Microbiologia, Parasitologia e Patologia, Departamento de Patologia Básica, Universidade Federal do Paraná (UFPR), Curitiba 81530-990, PR, Brazil; 4Laboratório de Bioinformática e Biologia de Sistemas, Pós-Graduação em Bioinformática, Universidade Federal do Paraná (UFPR), Curitiba 81520-260, PR, Brazil; 5Instituto de Pesquisa Pelé Pequeno Príncipe, Oncology Division, Curitiba 80250-060, PR, Brazil; 6Centro de Genética Molecular e Pesquisa do Câncer em Crianças (CEGEMPAC), Molecular Oncology Laboratory, Curitiba 80030-110, PR, Brazil; 7Inserm, 06560 Valbonne, France

**Keywords:** steroidogenic factor 1, transcription factors, nuclear receptors, adrenal cortex, steroidogenesis, gene dosage, tumorigenesis, adrenocortical carcinoma

## Abstract

Steroidogenic factor-1 (SF-1, also termed Ad4BP; NR5A1 in the official nomenclature) is a nuclear receptor transcription factor that plays a crucial role in the regulation of adrenal and gonadal development, function and maintenance. In addition to its classical role in regulating the expression of P450 steroid hydroxylases and other steroidogenic genes, involvement in other key processes such as cell survival/proliferation and cytoskeleton dynamics have also been highlighted for SF-1. SF-1 has a restricted pattern of expression, being expressed along the hypothalamic-pituitary axis and in steroidogenic organs since the time of their establishment. Reduced SF-1 expression affects proper gonadal and adrenal organogenesis and function. On the other hand, SF-1 overexpression is found in adrenocortical carcinoma and represents a prognostic marker for patients’ survival. This review is focused on the current knowledge about SF-1 and the crucial importance of its dosage for adrenal gland development and function, from its involvement in adrenal cortex formation to tumorigenesis. Overall, data converge towards SF-1 being a key player in the complex network of transcriptional regulation within the adrenal gland in a dosage-dependent manner.

## 1. Introduction

Adrenal glands are small paired organs located above each kidney, which participate in the coordination of the mammalian stress response alongside the hypothalamus and the pituitary gland [1]. They have a concentric structure, with a central medullary zone and an external cortical zone surrounded by a connective tissue capsule. Within the adrenal cortex, the outer *zona glomerulosa* (zG) produces mineralocorticoids, the central *zona fasciculata* (zF), glucocorticoids, and the inner *zona reticularis* (zR), androgens under the control of corticotrophin (ACTH) released by the pituitary. Both in humans and in mice, the structure of the adrenal cortex undergoes substantial remodeling postnatally.

Nuclear receptors (NR) represent one of the largest families of transcription factors, with approximately fifty members in mammals. They regulate essential biological processes by binding DNA and regulating the expression of target genes. NR family members are grouped into three classes further divided into seven subfamilies, according to their structural similarities [2]. The NR5A subfamily includes steroidogenic factor-1 (SF-1, also known as Ad4BP or, according to the official nomenclature, NR5A1) and liver receptor homolog-1 (LRH-1, NR5A2), which have different and mostly non-overlapping functions [3].

SF-1 was initially identified as a positive regulator of steroid P450 hydroxylases in adrenals and gonads [4,5,6], but has a broader role in the establishment and function of endocrine glands. Its presence in all hypothalamic–pituitary–steroidogenic organs makes it an essential regulator of various biological processes [3,6]. In addition to its role in steroidogenesis, recent genomic studies have demonstrated the implication of SF-1 in the processes of angiogenesis, cell adhesion, cytoskeletal dynamics, proliferation, apoptosis, and transcriptional and post-transcriptional regulation of gene expression [7]. Various studies have introduced the notion of SF-1 functional dosage, demonstrating that an adequate dosage of this transcription factor is essential for proper adrenal and gonadal development and function (reviewed in [8]). Interestingly, the phenotypes associated with altered SF-1 dosage depend on the species involved. While gonads in patients carrying heterozygote *NR5A1* mutations are most often affected by variable degrees of dysgenesis [9,10,11,12], haploinsufficiency of SF-1 causes adrenal hypoplasia and impaired function in mice [13,14]. In contrast, overexpression of SF-1 is associated with adrenocortical tumorigenesis in both species [8,15,16].

In this review, we have tried to provide a comprehensive overview of our current knowledge about the role of the transcription factor SF-1 in adrenal cortex physiopathology, with a special focus on its role in tumorigenesis.

## 2. SF-1 Structure

The SF-1 protein consists of four hundred and sixty-one, and four hundred sixty-two amino acids in humans and mice, respectively [6]. The gene (*NR5A1*; *Nr5a1*) encoding SF-1 is located on human chromosome 9 and mouse chromosome 2 [17], and consists of seven exons, the first being non-coding. SF-1 shares structural elements with other NRs (Figure 1).

Notably, its structure includes a DNA-binding domain (DBD) composed of two zinc fingers harboring a proximal (P) and a distal (D) box conferring binding affinity to consensus 5′-YCA**AGGYCR**-3′ sequences [18]. Importantly, NR5A transcription factors contain a C-terminal extension (CTE) after the zinc fingers region of the DBD that contacts the 5′-YCA-3′ portion of the binding site, allowing the protein to bind DNA as a monomer, similarly to a restricted group of other nuclear receptors. In addition, unique to the NR5A subfamily is another highly conserved sequence referred to as the “FTZ-F1 box”, located immediately C-terminal to the DBD CTE, which is required for transcriptional activity and interaction with coactivators. It contains a bipartite nuclear localization signal (NLS) [19].

The SF-1 N-terminal region upstream of the DBD is very short and does not possess a classical AF-1 domain, as observed in other nuclear receptors. The C-terminal part of the SF-1 protein contains a ligand-binding domain (LBD or E-domain) that includes a well-conserved AF-2 transactivation motif, which is also present in other ligand-dependent NRs [6,20]. In the case of SF-1, the presence of the AF-2 domain alone does not allow it to be defined as a ligand-activated nuclear receptor since its activity is constitutive [21]. Structural studies have shown that phospholipids can bind the pocket inside the SF-1 LBD stabilizing its fold, but it is still unknown whether they represent true physiological ligands [22,23,24]. On the other hand, the presence of this AF-2 domain is of crucial importance for SF-1 transcriptional activity, allowing the recruitment of transcriptional cofactors [20]. Between the DBD and the LBD, a hinge region referred to as domain D harboring a proline-rich domain is the target of post-translational modifications that modulate SF-1 activity [21,25].

## 3. SF-1 Expression

Spatial and temporal profiling of *Nr5a1* transcripts in mice has revealed their presence as early as embryonic day (E) 9 in the adrenogonadal primordium (AGP), the common structure from which the adrenal gland and the gonads are derived [26,27,28]. In humans, *NR5A1* transcripts are first detected in the urogenital ridge at 32 days post-ovulation (dpo)/Carnegie stage (CS) 14 [29], before adrenal and gonadal differentiation. In both species, SF-1 expression precedes that of the testis-determining *SRY* [6,29].

At E12.5 (mouse) or 44 dpo/CS18 (humans), gonadal differentiation begins with the formation of testicular cords and in mice is accompanied by sexual dimorphism of *Nr5a1* expression. In fact, in this species, *Nr5a1* transcripts are detected throughout the development of the testis, whereas in the ovary it is not detected between E13.5 and E16.5 [6,26,27]. Conversely, in humans, *NR5A1* transcripts are detectable in the ovary throughout its development [12,29]. In mice, variations in *Nr5a1* expression have also been observed during adrenal development. After the formation of the gonadal (AG) and adrenal primordia (AP) from the AGP between E10.5 and E11, *Nr5a1* continues to be expressed in adrenal progenitor cells [30]. However, after separation of the adrenal cortex and medulla, between E18.5 and postnatal day (PND) 6, its expression transiently decreases. At this time and until adulthood, SF-1 expression is restricted to the cortex [31]. In adult mice, SF-1 has restricted and specific expression patterns in the hypothalamic–pituitary–steroidogenic axis [26,32]. In female mice, it is expressed in granulosa and theca cells starting from the beginning of ovarian folliculogenesis [32,33] and, later on, in the corpus luteum [4]. Its presence has also been detected in ovarian interstitial cells [34,35]. In male mice, it is found both in Leydig and in Sertoli cells [26,36]. In mouse tissues, other than in the gonads and steroidogenic cells of the adrenal cortex, SF-1 is expressed in ventromedial hypothalamus [26,37,38] and gonadotropes of the pituitary gland [39,40,41] (Figure 2). In addition, it is expressed in a subset of hippocampal neurons [42], the spleen and its veinous sinuses [43].

Given its function as transcription factor, SF-1 is localized in the cell nucleus [3]. It has also been shown to colocalize with the centrosome marker γ-tubulin in mouse MA-10 Leydig and Y1 adrenocortical cells. Deletion analysis identified a centrosome localization signal at aa 348-367, which may be of importance during cell division to prevent centrosome overduplication and genomic instability [44,45].

## 4. A Key Role for SF-1 in Adrenal Gland Organogenesis

Adrenal gland formation is a complex process and even if differences exist between humans and mice in its organogenesis, several genes involved are common to both species [46,47]. Briefly, once the AP is formed from the AGP, the adrenocortical lineage forms morphologically and functionally distinct areas: the fetal inner area adjacent to the medulla (termed X-zone in mice) and the definitive outer area [48]. The fetal cortex is gradually replaced by the definitive cortex and the encapsulation of the fetal cortex marks the beginning of the transition. At that time, the different zones of the cortex are not established yet. This event, which involves cell differentiation, occurs during early postnatal life. The maintenance of those zones is then ensured by a constant renewal process throughout life [49]. During embryogenesis in humans and in some primate species, the adrenal cortex is composed in the largest extent of large cells producing adrenal androgens [mainly dehydroepiandrosterone sulfate (DHEAS)] (fetal zone: FZ), separated by a transitional zone (TZ) from an outer layer of smaller cells (definitive zone: DZ), which contains the adrenal precursor cells. After birth, the FZ undergoes rapid involution, while the DZ expands and differentiates to form the characteristic zones of the adult cortex zG, zF, and later on, zR [50]. Many of the processes occurring during adrenal formation involve SF-1, from organogenesis to maintenance in the adult [49]. Different mouse models as well as high throughput genomic analyses have expanded our knowledge about SF-1 function, which far exceeds its classic function in the regulation of genes involved in steroidogenesis [7,31]. An essential role for SF-1 in adrenal gland development was demonstrated using knock-out mice models for *Nr5a1* that lack adrenals and gonads at birth [36,51]. In those animals, adrenal glands and gonads initiated a normal developmental program, however since E12, the AP and AG regressed due to apoptosis and pups died because of adrenal insufficiency [36]. As mentioned above, the first SF-1 expression during adrenal development is found at E9 in mice and at 30 dpc in humans [26,27,29]. These stages of embryonic development coincide with the establishment of the adrenal gland and gonadal precursor cells. The understanding of the mechanisms underlying adrenal development in humans is still far from exhaustive [52]. While both the adrenal cortex and gonads derive from the AGP in rodents, that originates from the thickening of the coelomic epithelium at E9 [53], it has recently been shown that in humans and cynomolgus monkeys the adrenal and gonadal precursors arise from opposite regions of the coelomic epithelium and at distinct developmental stages, respectively anterior at 30 dpc and posterior at 33 dpc [47]. Despite the differences in the emergence of adrenal precursor cells in these two species, a high expression of SF-1 is a common element. Current data suggest that SF-1 in these adrenal precursor cells is important to initiate the steroidogenic cell differentiation program. Additionally, exogenous SF-1 expression in mouse embryonic stem cells has been shown to trigger the expression of *Cyp11a1*, a key enzyme involved in the process of steroidogenesis, as well as progesterone synthesis in response to cAMP [54]. Further works on the differentiation of mesenchymal [55], murine [56,57] or human [58] bone marrow, adipose tissue [59] or umbilical cord [60] stem cells, have demonstrated a key role for SF-1 in establishing a differentiation program into steroidogenic cells. Similarly, human induced steroidogenic cells (hiSCs) were generated from fibroblasts, blood-, and urine-derived cells through forced expression of SF-1 and activation of the PKA pathway [61].

In mice, SF-1 expression is induced shortly after that of *Gata4* and *Wt1*, which have an important role in the formation of the AGP [28,30]. Subsequently, through the action of homeobox proteins and positive feedback involving the fetal adrenal enhancer (FadE) located in the fourth intron of the *Nr5a1* gene, SF-1 expression is specifically increased in the AP [62]. An in vivo model using the FadE to induce overexpression of Ad4BP/SF-1 during mouse development supported the idea that not only SF-1 affects steroidogenic cell fate determination, but also that its precise regulation is essential for proper adrenal development. In this model of SF-1 overexpression, ectopic adrenal tissue formation in the thorax was reported, as well as an increase in adrenal size without an impact on cell proliferation [63]. The presence of such a regulatory element in the human gene sequence has not yet been demonstrated, although the FadE locus shares a high degree of homology in both species [62]. The interaction between SF-1 and DAX-1, a negative regulator of SF-1 activity [64], in synergy with the SUMOylation of SF-1 (see below), is required to extinguish FadE activation in the postnatal fetal cortex, which then undergoes regression [65,66,67]. This is consistent with data showing that the X zone is maintained in adult male *Dax1*^-/y^ mice [68].

In addition to its role in cell fate determination, SF-1 is required for adrenocortical cell proliferation. SF-1 overexpression leads to increased cell proliferation in the mouse definitive adrenal cortex and in the H295R adrenocortical tumor cell line [8,15]. Conversely, *Nr5a1*^+/−^ mouse embryos exhibit hypoplastic adrenals resulting in adrenal insufficiency and, therefore, altered stress response [13]. In the *Nr5a1* knock-out mouse model, exogenous expression of Ad4BP/SF-1 rescued the gonadal but not the adrenal defects [69], indicating that an altered SF-1 dosage severely affects adrenal development in mice. Interestingly, when the alteration of SF-1 expression is a consequence of the deletion of genes such as *Cited2*, *Wt1* [28], *M33* [70], or *Fgfr2b* [71], adrenal development is impaired (Table 1). This emphasizes the major role of SF-1 in the development of the adrenal gland and indicates that not only the presence of SF-1 but also its adequate levels are required for proper adrenal development.

## 5. Regulation of SF-1 Expression and Activity in Adrenal Glands

Regulation of both SF-1 expression and activity appears to be crucial as altered functional dosage results in disturbed adrenal and gonadal development and function, causing different pathologies.

The *Nr5a1* gene is regulated by multiple tissue-specific enhancers [75]. As mentioned above, overexpression of SF-1 using the FadE results in a forced commitment to a steroidogenic fate [63]. Regarding other organs, a VMH-specific enhancer [76] and a pituitary enhancer [77], both localized in the sixth intron of the mouse *Nr5a1* gene locus, have been identified that allow localization of SF-1 expression to the respective tissues. Further, tissue-specific regulation of SF-1 may be achieved by interacting proteins which positively or negatively regulate its activity, and are themselves expressed in a tissue-specific manner. A relevant example is DAX-1 [78], which was introduced above. Its expression pattern closely follows that of SF-1, and phenotypes resulting from an impaired DAX-1 expression resemble those induced by an altered dosage of SF-1, as reviewed in [79]. DAX-1 inhibits SF-1-dependent gene expression, either through direct binding to SF-1 [64], or by binding to the promoter region of its target genes [80]. In addition to tissue specificity, the timing of expression seems to control SF-1 activity and, therefore, adrenal development. In an attempt to rescue adrenal development in SF-1 null mice, BAC transgenesis was used. Adrenal glands failed to develop, while the spleen and the gonad developed normally [69]. Early SF-1 expression in the adrenogonadal primordium is under the control of *Wt1* and *Cited2* [28]; *Wt1*, in turn, being regulated by *Odd1* [81] and *Sall1* [82]. MicroRNAs, which are small non-coding RNAs controlling gene expression, might also regulate SF-1 at the post-translational level. This type of regulation remains poorly explored, however several elements suggest that they might limit SF-1 expression. The *NR5A1* 3′UTR of different vertebrate species harbors a predicted miR125b binding site. This miRNA is significantly downregulated in pediatric adrenocortical tumors (ACT) [7,83] and may be implicated in the mechanism of SF-1 upregulation in those tumors, especially in cases lacking *NR5A1* copy number gain (see below).

At the post-translational level, SF-1 activity is regulated by SUMOylation. This transcription factor is constitutively targeted to the nucleus, however, because of this post-translational modification, its nuclear distribution is modified. The main SUMOylation site of SF-1 is Lys194. This modification is associated with repression of its activity by recruitment of negative regulators, such as the DEAD-box protein DP103. The localization of SUMOylated SF-1 is restricted to nuclear speckles, thus, preventing its DNA-binding activity [65,84]. Conversely, the localization of SF-1 in active foci within the nucleus, which allows for transcription of target genes, has been reported to be under the control of acetylation at the KQQKK motif in the FTZ-F1 box and of the cAMP pathway [85]. The acetylation state of SF-1 has a positive impact on its half-life and activity [86]. In addition, in Y1 mouse adrenocortical cells, cAMP stimulated the expression of the histone acetyltransferase p300, which associated with SF-1, increases its acetylation and DNA binding [85]. Thereby, a cAMP-dependent mechanism has been thought to promote interactions between SF-1 and cofactors [85,87,88] and to induce the synthesis of activating ligands for SF-1 [89,90]. Other post-translational modifications such as phosphorylation at Ser203 [25] or ubiquitination [91] do not affect SF-1 subcellular localization or its binding to DNA, however Ser203 phosphorylation is necessary to achieve SF-1 maximal activity [25]. Two kinases are known to phosphorylate SF-1, ERK2 and CDK7 [25,92,93]. CDK7-induced phosphorylation of SF-1 can be inhibited by its SUMOylation [94,95].

A crucial factor regulating SF-1 expression and biological activity is represented by its dosage. The significance of SF-1 gene dosage has already been introduced above with regard to the adrenal defects in mice resulting from altered SF-1 expression. In general, any altered SF-1 dosage leads to non-physiological conditions both in mice and humans. In the case of heterozygous mutations of *NR5A1*, a wide range of phenotypes is observed in patients: both gonadal and, more uncommonly, adrenal development and function may be affected, which leads to different degrees of gonadal dysgenesis, male infertility, primary ovarian and adrenal insufficiency [96]. Reported cases of *NR5A1* mutations causing adrenal failure are the G35E (heterozygote), R92Q (homozygote) and R255L (heterozygote) missense mutations affecting DNA binding. Among those cases, only the R255L mutation selectively affected adrenal function without producing a gonadal phenotype [9,96,97].

## 6. SF-1 Dosage Is not Only Critical for Adrenal Development but also for Tumorigenesis

Analysis of both human and murine adrenal tissues has linked SF-1 overexpression to adrenocortical tumorigenesis, providing insights into the underlying mechanisms of tumor formation (reviewed in [8]). An intriguing aspect comes from the study of a group of pediatric ACT from southern Brazil, where the incidence of this malignancy is remarkably higher than in the rest of the world because of the high prevalence of a specific *TP53* germline mutation (p.R337H) in the population [98]. In this cohort, frequent amplification of the 9q33–q34 region, in which the *NR5A1* gene is located, was found [99]. Later on, increases in *NR5A1* copy number and SF-1 protein overexpression were demonstrated in most cases of pediatric ACT [100,101,102]. High levels of SF-1 were also reported in adrenocortical carcinoma (ACC) in adults and negatively correlated to patients’ overall survival (OS) [16].

In an animal model, transgenic mice harboring YAC clones containing several copies of the rat *Nr5a1* gene developed adrenocortical neoplastic lesions (both dysplastic and nodular) with a frequency increasing with age [15,103]. In these animals, tumors arose in the adrenal subcapsular region and expressed the gonadal markers Gata4 and AMH, suggesting that tumor cells are derived from undifferentiated adrenogonadal precursors [15]. The critical role of SF-1 in adrenocortical tumorigenesis is reinforced by data from a human adrenocortical cell line (H295R) where SF-1 can be overexpressed in a doxycycline-inducible manner. SF-1 overexpression in H295R cells is able to sharpen their malignant phenotype, increasing their proliferation and invasive capacity in vitro and decreasing apoptosis [15,104]. This effect is dependent on SF-1 transcriptional activity, since overexpression of an AF-2 SF-1 mutant has no effect on cell proliferation. In a concordant fashion, in that cell model, changes in steroid hormone biosynthesis induced by SF-1 overexpression recapitulated those present in pediatric ACT, which produce high DHEAS and lower aldosterone and cortisol levels [15]. Furthermore, an SF-1 synthetic inverse agonist prevents the increase in adrenocortical cell proliferation and the changes in hormone production induced by SF-1 overexpression in H295R cells, indicating that this transcription factor is a druggable target in ACC [105].

## 7. SF-1 Dosage-Dependent Target Genes: Their Roles in ACC and in Adrenal Gland Development

It is remarkable that SF-1 overexpression in the H295R cell model was sufficient to induce modifications of gene expression which are consistent with patterns found in pediatric ACT, modulating the expression of genes related to cell cycle, apoptosis, cell adhesion and steroidogenesis [15,106]. ChIP-seq studies coupled to gene expression profiling have shown that SF-1 regulates distinct, mostly non-overlapping sets of target genes in ACC cells according to its dosage [7,107]. Functional studies about the molecular and cellular roles of some among those SF-1 dosage-dependent target genes have elucidated their important roles in adrenocortical tumorigenesis:-*NOV/CCN3* encodes a secreted protein with pro-apoptotic properties which is a marker of the DZ in the human fetal adrenal cortex. Its expression is reduced in ACC. SF-1 overexpression in H295R cells significantly reduces NOV expression [108];-*FATE1* encodes a protein enriched in mitochondria-associated membranes (MAM) which has an important role in modulating calcium transfer between the endoplasmic reticulum and mitochondria in ACC cells, being involved in the resistance to chemotherapeutic drugs [109,110]. In addition, FATE1 is a prognostic factor in ACC and a cancer-testis antigen against which an immune response is present in patients with ACC [111], making it a potential target for immunotherapy;-*VAV2* encodes a guanine nucleotide exchange factor (GEF) which activates Rho family GTPases, important regulators of cell cytoskeleton, motility and invasion. VAV2 is a prognostic factor in ACC and its knockdown significantly inhibits H295R cell invasion in Matrigel activated by SF-1 overexpression [104].

Altogether, these findings support a ‘Goldilocks’ model of transcriptional regulation by SF-1, in which the levels of active transcription factor protein must be ‘just right’ to direct transcriptional regulation of its physiological target genes [112]. Decrease or increase in SF-1 activity/levels below or in excess of a critical threshold leads to defects in adrenal gland development or is associated with adrenal tumorigenesis, respectively (Figure 3).

These properties of SF-1 may also have an important role during human adrenal development: it has been reported, in fact, that higher levels of SF-1 are expressed in the FZ compared with the DZ [47], which correlates to the suppression of genes involved in the production of cortisol [15] and to the reduced expression of DZ marker genes (*NOV/CCN3*) induced by an increased dosage of SF-1 in the H295R cell model [108]. It is then tempting to speculate that changes in SF-1 expression and/or activity have an important role in the modulation of adrenocortical cell differentiation towards a FZ vs. DZ phenotype. Interestingly, the expression pattern of the DAX-1 repressor in the fetal adrenal is the opposite of SF-1, with higher expression and more pronounced nuclear localization in the DZ compared with the FZ [113]. In this model, lower SF-1 activity in the DZ would allow for progenitor cell maintenance and subsequent differentiation into cells of the definitive cortex, while higher SF-1 activity in the FZ is associated with their androgenic steroid secretion pattern (Figure 4).

The identification of upstream factors and signaling pathways finely tuning the differential expression of SF-1 in the different cell layers of the human fetal adrenal gland is then of utmost importance for future studies. This may be made easier by the recent development of an in vitro system to produce human fetal adrenal cells from induced pluripotent stem cells (iPS) [114].

## 8. SF-1 and the Tumor Microenvironment (TME) in ACC

Homogeneous culture conditions obtained from patient-derived cell lines are a simple and low-cost tool to study druggable targets and mechanisms. They are widely used in pre-clinical studies, where many authors have identified drugs that can potentially be used in clinical trials [115]. However, gene expression profiles may present differences in tumor cell lines compared with tumor samples. Data from RNA expression profiling in tumors represent a genetic mosaic, displaying heterogeneous gene expression patterns for each cell type, containing reads from blood, stroma and immune infiltrate, beyond tumor cells themselves. When bulk RNA-seq data from tumors are considered, network analysis can exploit this mixture of gene expression profiles from different cell types. Therefore, the concept of regulatory networks based on transcription factors (TF), as described elsewhere [116,117], takes into consideration target genes either directly or indirectly stimulated or inhibited by the TF.

A great deal of effort has been focused on defining the role of TFs associated with the pathogenesis and/or prognosis of ACC. Those studies have been made possible by accessing publicly available transcriptome databases to evaluate the intratumoral microenvironment. Using the RNA-seq and clinical data from 78 samples from the TCGA ACC cohort [118], Muzzi et al. identified 369 regulatory units composed of a TF and its targets, named regulons, associated with OS in multivariate Cox analysis [119]. Of these 369 units, 346 were also significantly correlated to OS in the European Network for the Study of Adrenal Tumors (ENSAT) cohort with 44 ACC cases [120]. The NR5A1 regulon represented one of the largest regulons in this analysis with 248 targets [119]. Overexpression of this TF correlated with low OS in the multivariate Cox and Kaplan–Meier analyses in the TCGA ACC cohort as well as in the ENSAT ACC cohort. From the regulatory network analysis, the NR5A1 regulon activity was found to be related to worse outcomes in OS and progression-free interval both in the TCGA-ACC cohort, as well as in the ENSAT cohort. Interestingly, of the 369 prognostic regulons, the NR5A1 regulon presented the highest negative correlation with the TGF beta response, and a negative correlation with IFN gamma response, lymphocyte infiltration signature, and leukocyte fraction. On the other hand, it was positively correlated with proliferation and wound healing signatures [119]. These results indicate an association of NR5A1 with immune suppression observed in most ACC cases [121,122] and may be related to the stimulation of steroidogenesis by this TF. However, a full understanding of the role of NR5A1 in the immune response in ACC still has to be obtained. Interestingly, the regulatory network analysis revealed genes that may be indirect targets of SF-1, not just in tumor cells but also in other cell types in the ACC TME.

## 9. Conclusions and Future Perspectives

Since its discovery, the SF-1 transcription factor has generated wide interest, which has led to its designation as a “master regulator” of major endocrine organ development and physiology. A crucial role in the cellular processes allowing the establishment of the gonads and adrenals, their development and function has been widely demonstrated for SF-1. In particular, adrenal gland formation, maintenance, steroidogenesis and tumorigenesis are dependent on the tight regulation of SF-1 dosage in a time- and tissue-specific manner. Further studies are needed to improve our understanding of the mechanisms regulating both the expression and the biological activity of SF-1, which would allow us to better understand and counteract the pathological consequences of the alterations of its functional dosage.

## Figures and Tables

**Figure 1 ijms-24-03585-f001:**
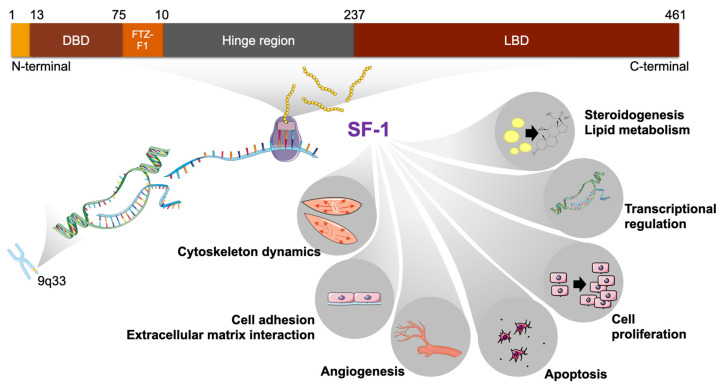
Structure of the human SF-1 protein and its target gene categories in tumor adrenocortical cells. The protein domains are shown and corresponding amino acid positions are indicated. SF-1 regulates several key cellular processes. DBD, DNA binding domain; FTZ-F1, FTZ-F1 box; LBD, ligand binding domain.

**Figure 2 ijms-24-03585-f002:**
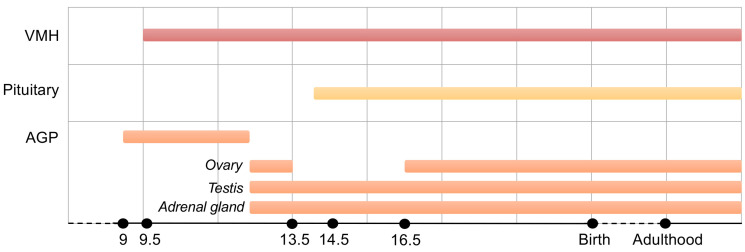
Timing of SF-1 expression in the mouse embryo along the hypothalamic–pituitary–adrenogonadal axis. SF-1 is expressed in those organs starting from the early stages of development, in particular in the AGP. The developmental stages before birth are indicated in days post-conception.

**Figure 3 ijms-24-03585-f003:**
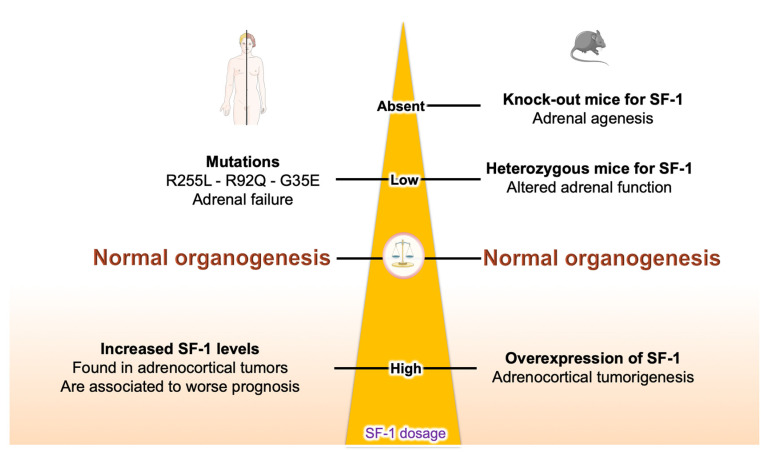
Adrenal phenotypes associated with an altered SF-1 dosage in mice and humans.

**Figure 4 ijms-24-03585-f004:**
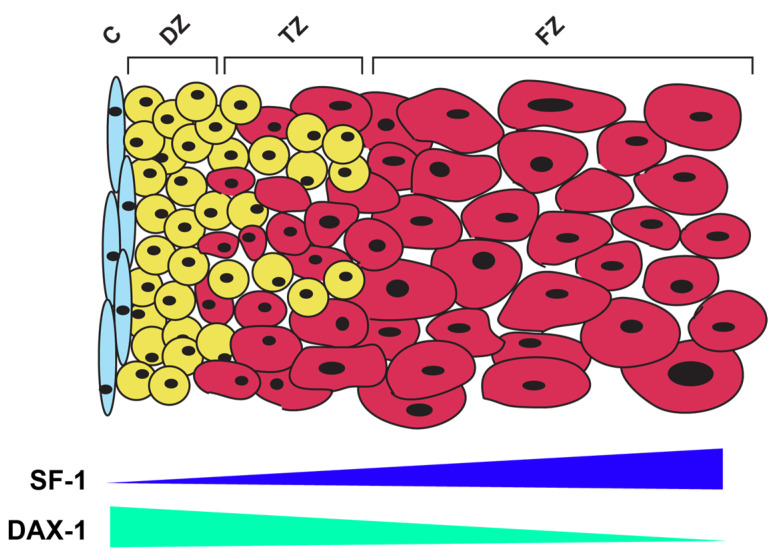
A model for the role of SF-1 in the control of adrenocortical cell differentiation in the human fetal adrenal. A gradient of SF-1 expression exists, with higher levels in the FZ compared with the DZ. Additionally, the DAX-1 repressor is expressed at higher levels and shows more pronounced nuclear localization in the DZ compared with the FZ. This leads to presumed higher transcriptional activity of SF-1 in the FZ compared with the DZ and differential regulation of the expression of its dosage-dependent target genes. C, adrenal capsule; DZ, definitive zone; TZ, transitional zone; FZ, fetal zone.

**Table 1 ijms-24-03585-t001:** Mouse models with altered SF-1 expression.

Mouse Model	Adrenal Phenotype	Observed in the Model	Reference
*Cited2* ^−^ ^/−^	Absence of adrenal glands	Lack of Sf-1 expression	[28]
*Cited2*^+/−^; Wt1^+/−^	Marked decrease in adrenal size at E11.5	Decrease in Sf-1 expression due to activation of *Nr5a1* promoter by Cited2 et Wt1	[28]
*Six1*^−/−^; *Six4*^−/−^	Smaller adrenal glands than wild type counterparts	Reduced *Nr5a1* expression due to the ability of Six1 and Six4 to bind to the *Nr5a1* promoter	[72]
*M33* ^−^ ^/−^	Underdeveloped adrenal glands	Decreased *Nr5a1* expression through indirect binding to the *Nr5a1* locus	[70]
*Insr*^−/−^; *Igf1r*^−/−^	Adrenal agenesis for the majority of mutant animals; drastic reduction in adrenal size for a small number of mutants	Altered expression of the core AGP program including a reduction in *Nr5a1* transcript levels	[73]
*Pbx1* ^−^ ^/−^	Absence of adrenal glands	Decrease in SF-1 expression in *Pbx1*^-/-^ embryos correlated with the absence of the adrenocortical cell population	[74]
*Fgfr2b* ^−^ * ^/^ * ^−^	Adrenal hypoplasia	Lack of *FGFR2b* splice variant which causes reduced *Sf-1* mRNA levels and reduction in inner cortical cell proliferation	[71]
*Sf1^2KR^* ^/2KR^	Smaller adrenal glands and delayed postnatal regression of adrenal X-zone	SUMOylation-deficiency leading to persistent FadE activity in X-zone	[66,67]
*Dax1* ^−/y^	Delayed postnatal regression of adrenal X-zone	Dax1-deficiency leading to persistent FadE activity in X-zone due to its repressor role towards Sf-1	[67]

## Data Availability

No new data were created or analyzed in this study. Data sharing is not applicable to this article.

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
