# Peer review of "Steroidogenic Factor 1, a Goldilocks Transcription Factor from Adrenocortical Organogenesis to Malignancy"

_ijms, 2023, doi:10.3390/ijms24043585_

Round 1

Reviewer 1 Report

The review entitled “Steroidogenic Factor 1, a Goldilocks transcription factor from adrenocortical organogenesis to malignancy” by Lalli and coauthors is a comprehensive and upgraded manuscript on the pleyotropic functions of SF-1 in adrenal gland.

The manuscript  is well written, extensively examining the physiological and pathological roles of SF-1 from embryonic to adult development.

I think that the Ms deserves publication in IJMS, with minor revisions.

In particular, in some points,  Authors report data and draw conclusions for which it is not clear whether they are drawn from the review of results obtained in experimental animals (rat or mouse) or in humans. As an example (but not limited to): paragraph 3, line 118, line 121. I would suggest to go through  the whole manuscript and clarifying these aspects.

Further, please correct  typos and misspelling throughout the manuscript

Author Response

Thank you for your comments on our manuscript. We have modified the text according to your suggestions in paragraph 3 and throughout the manuscript. We have also corrected all the typos and misspellings.

Reviewer 2 Report

Since this manuscript is interesting and contains important aspects, I think it O.K. to accept it.

Author Response

We thank this reviewer for her/his comments.

Reviewer 3 Report

IJMS

COMMENTS TO THE EDITORS AND THE AUTHORS

ijms-2183808: “Steroidogenic Factor 1, a Goldilocks transcription factor from adrenocortical organogenesis to malignancy”

Dear the Editors and the Authors,

Please find enclosed the comments for the above-mentioned manuscript.

A SUMMARY OF THE CONTENT

The authors stated that review is focused on the current knowledge about SF-1 and the crucial importance of its dosage for adrenal gland development and function, from its involvement in adrenal cortex formation to tumorigenesis. The authors concluded that results suggest SF-1 being a key player in the complex network of transcriptional regulation within the adrenal gland in a dosage-dependent manner.

THE OVERALL OPINION OF THE MANUSCRIPT

The strengths. 

The manuscript is within the scope of the journal and addresses the important question. The objectives and justification are very clearly stated. The authors presented nicely knowledge in the field.

The limitations.

The citation and the discussion of the original and the important pioneered results in the field focusing on the subject of the study is missing; the discussion of the sex-dependent observation is missing.

SUGGESTIONS

(1) Please focus the text of the abstract on the aim and the results. Namely, the text of the abstract dedicates only 5 out of 13 rows to the aim and the results. 

(2) Please describe and the discuss the original and the important pioneered results in the field focusing on the subject of the study is missing.

(3) Please precisely describe the source of the results (type and sex of the cells, tissue etc.).

(4) Please discuss the sex-dependent differences.

(5) Please consider adding one paragraph describing the future perspectives.

Accordingly, minor revision is required.

I would greatly appreciate if you will contact me if you find something in my comments is missing/unclear/incorrect.

Good luck and all the best J

Author Response

We thank this reviewer for her/his comments. Here is our rebuttal to the points s/he raised:

1) Since this is a review article, we believe it is justified that the abstract summarizes the main points of our knowledge of what is known about the SF-1 transcription factor.

2) We feel that in our review we have extensively described all the milestone work in the field, especially the studies performed in Dr. Morohashi's and Dr. Parker's laboratories. We cited the following references from those groups:

- Morohashi’s lab: 5, 27, 33, 37, 43, 48, 62, 63, 69, 70, 72, 75, 76, 77

- Parker’s lab: 4, 6, 26, 29, 32, 36, 39, 78

3) We have modified Paragraph 3 and all the other text to take into account this comment (please also see our rebuttal to Reviewer 1).

4) See point 3).

5) We have modified the last paragraph as "Conclusion and future perspectives".